# Optical control of pain *in vivo* with a photoactive mGlu5 receptor negative allosteric modulator

Joan Font[1,2,3†], Marc López-Cano[4,5†], Serena Notartomaso[6], Pamela Scarselli[6], Paola Di Pietro[6], Roger Bresolí-Obach[7], Giuseppe Battaglia[6], Fanny Malhaire[8], Xavier Rovira[8], Juanlo Catena[1], Jesús Giraldo[2,3,9], Jean-Philippe Pin[8], Víctor Fernández-Dueñas[4,5], Cyril Goudet[8], Santi Nonell[7], Ferdinando Nicoletti[6,10], Amadeu Llebaria[1]*, Francisco Ciruela[4,5]*

[1]MCS, Laboratory of Medicinal Chemistry, Institute for Advanced Chemistry of Catalonia (IQAC-CSIC), Barcelona, Spain; [2]Institut de Neurociències, Universitat Autònoma de Barcelona, Bellaterra, Spain; [3]Unitat de Bioestadística, Universitat Autònoma de Barcelona, Bellaterra, Spain; [4]Departament de Patologia i Terapèutica Experimental, Facultat de Medicina i Ciències de la Salut, IDIBELL, Universitat de Barcelona, Barcelona, Spain; [5]Institut de Neurociències, Universitat de Barcelona, Barcelona, Spain; [6]I.R.C.C.S. Neuromed, Pozzilli, Italy; [7]Institut Químic de Sarrià, Universitat Ramon Llull, Barcelona, Spain; [8]IGF, CNRS, INSERM, Univ. Montpellier, Montpellier, France; [9]Network Biomedical Research Center on Mental Health (CIBERSAM), Instituto de Salud Carlos III, Madrid, Spain; [10]Department of Physiology and Pharmacology, University Sapienza, Rome, Italy

*For correspondence: amadeu.
llebaria@iqac.csic.es (AL);
fciruela@ub.edu (FC)

†These authors contributed
equally to this work

Competing interests: The
authors declare that no
competing interests exist.

Reviewing editor: Gary L
Westbrook, Vollum Institute,
United States

**Abstract** Light-operated drugs constitute a major target in drug discovery, since they may provide spatiotemporal resolution for the treatment of complex diseases (i.e. chronic pain). JF-NP-26 is an inactive photocaged derivative of the metabotropic glutamate type 5 (mGlu5) receptor negative allosteric modulator raseglurant. Violet light illumination of JF-NP-26 induces a photochemical reaction prompting the active-drug's release, which effectively controls mGlu5 receptor activity both in ectopic expressing systems and in striatal primary neurons. Systemic administration in mice followed by local light-emitting diode (LED)-based illumination, either of the thalamus or the peripheral tissues, induced JF-NP-26-mediated light-dependent analgesia both in neuropathic and in acute/tonic inflammatory pain models. These data offer the first example of optical control of analgesia *in vivo* using a photocaged mGlu5 receptor negative allosteric modulator. This approach shows potential for precisely targeting, in time and space, endogenous receptors, which may allow a better management of difficult-to-treat disorders.

## Introduction

Metabotropic glutamate (mGlu) receptors are widely distributed along the pain neuraxis and modulate pain transmission (*Kolber, 2015*) at different anatomical levels. Hence, subtype-selective mGlu receptor ligands are considered as promising candidate drugs for the treatment of chronic pain. Accordingly, selective negative allosteric modulators (NAMs) of mGlu1 or mGlu5 receptors, and agonists or positive allosteric modulators (PAMs) of mGlu2 or mGlu4 receptors have consistently been shown to display analgesic activity in experimental animal models of chronic pain (*Montana and Gereau, 2011*). In particular, mGlu5 receptor NAMs (i.e. raseglurant) are under development for the

treatment of neuropathic pain and migraine; however, their systemic use may be limited by mechanism-related adverse effects, such as hepatotoxicity, cognitive impairment and psychotomimetic effects (*Friedmann et al., 1980*; *Gravius et al., 2005*; *Homayoun et al., 2004*; *Kinney et al., 2003*).

The use of light in optogenetics (*Tye and Deisseroth, 2012*) and photopharmacology (*Kramer et al., 2013*; *Lerch et al., 2016*) to control activity of living neurons is a ground-breaking approach that is reaching a high impact in experimental neuroscience (*Boyden, 2015*) and drug discovery (*Song and Knöpfel, 2016*). Photopharmacology is based on the use of photoactive drugs that can trigger biological responses upon light irradiation at appropriate wavelengths. Accordingly, temporal and spatial remote control of drug activity is possible with a highly precise regulation of its in vivo effects. Differently from optogenetics, the application of photoregulated ligands and optical techniques allows the modulation of native proteins without genetic manipulation, thus facilitating translation into therapeutics with protocols similar to those employed in conventional drug discovery and clinical practice. The potential application of this strategy to pain treatment is highlighted by the evidence that a photoisomerizable molecule entering nociceptive neurons through the TrpV1 ion channel causes analgesia by inhibiting voltage-gate ion channels in response to illumnation (*Mourot et al., 2012*). We developed alloswitch-1, a selective photoswitchable mGlu$_5$ receptor NAM, which allowed the optical control of endogenous receptors (*Pittolo et al., 2014*), in an attempt to achieve a light-dependent control of mGlu$_5$ receptors within the pain neuraxis. However, alloswitch-1 and related photoswitchable compounds show mGlu$_5$ NAM activity in its natural *trans* configuration in the dark, thereby causing light-independent analgesic effects in animals (*Gómez-Santacana et al., 2017*). Therefore, we aimed to synthesize an inactive mGlu$_5$ receptor NAM, which is converted into an effective analgesic drug upon illumination. An effective strategy to achieve this would consist of developing inactive photosensitive mGlu$_5$ receptor-based drugs, which may thereafter be activated by light administration (photo-uncaging) exclusively in brain regions critically involved in the control of pain. This approach may allow the release of the active drug in an anatomically restricted fashion and with regulated dosing. The photodelivery of biologically active molecules has been usually addressed using either direct compound photolysis (*Ellis-Davies, 2007*) or light-triggered release from nanosystems (*Fomina et al., 2012*), although uncaging in vivo has been seldom used in rodents (*Crowe et al., 2010*; *Takano et al., 2006*). Here, we aimed to develop a photoactivatable mGlu$_5$ receptor NAM, which upon local illumination in the hind paw or the thalamus would exert analgesic effects.

## Results

Design and synthesis of a caged mGlu$_5$ receptor NAM. We applied a caging strategy to generate a photocaged compound based on the chemical binding of the mGlu$_5$ receptor NAM, raseglurant (ADX-10059), to a coumarinyl phototrigger (*Figure 1*). Thus, we synthesized JF-NP-26 (*Figure 1*) by modifying the aromatic amine group in raseglurant to generate a carbamate derivative of the violet-light absorbing coumarin DEACM in a one pot procedure (*Figure 1*).

Transformation of the amino group of raseglurant into the coumarinyl carbamate shifted its main UV-vis absorption band from 338 nm to around 313 nm, and a second absorption peak appeared at 386 nm due to the coumarinyl phototrigger (*Figure 2A*). Subsequently, we assessed the photochemical behaviour of JF-NP-26 upon exposure to 405 nm light (*Figure 2B*), which is safer than UV radiation for our in vivo biological experiments. The UV-visible absorption spectrum of JF-NP-26 showed dramatic changes upon 405 nm irradiation (*Figure 2C*), consistent with the photo-induced release of raseglurant and leaving the coumarinyl alcohol DEACM (*Figure 2A*). The photouncaging quantum yield was determined by potassium ferrioxalate actinometry (*Kuhn et al., 1989*), as $\varphi_{chem}$ = 0.18 (*Figure 2D*). This process is likely involving the photolysis of the benzylic bond in JF-NP-26, thus generating a carbamic acid derivative that spontaneously decarboxylates to give raseglurant.

## Optical modulation of mGlu$_5$ receptor activity with JF-NP-26 in cultured cells and in primary neurons

Subsequently, we assessed the JF-NP-26-mediated negative allosteric modulation of mGlu$_5$ receptor-induced responses to the orthosteric agonist quisqualate, by using an inositol phosphate (IP) accumulation assay (*Figure 3A*). Interestingly, while JF-NP-26 didn't show activity in dark conditions, its NAM activity was rescued upon 405 nm visible light illumination (pIC$_{50}$ = 7.1; *Figure 3A*). Thus, a

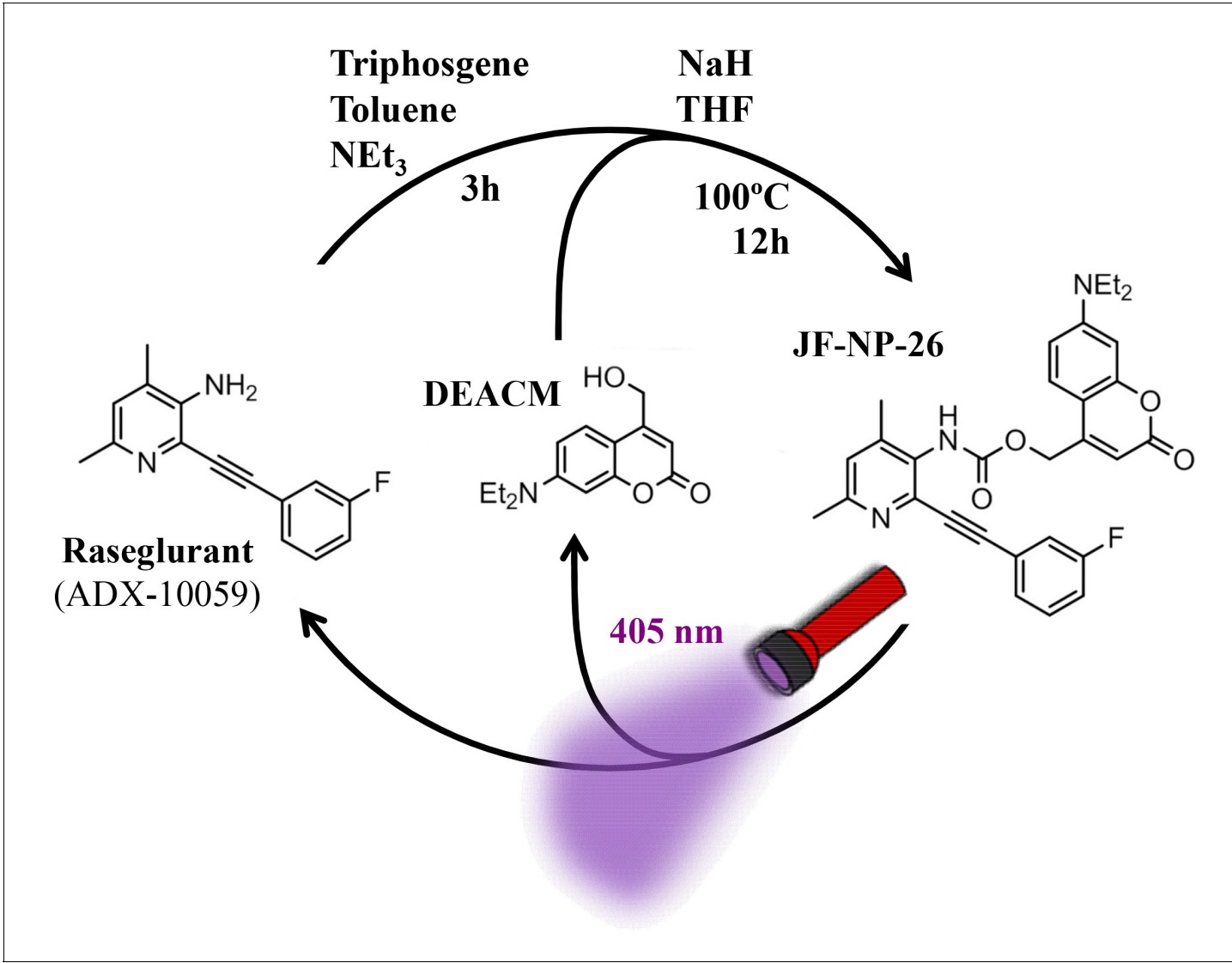

**Figure 1.** Design and synthesis of JF-NP-26. The synthesis of JF-NP-26 from raseglurant involves a one-pot procedure using raseglurant and 4-hydroxymethyl-7-diethylaminocoumarin (DEACM). In brief, a first reaction with triphosgene, $NEt_3$ and toluene for 3 hr was followed by an incubation with DEACM, NaH and THF at 100°C for 12 hr (see Materials and methods). Upon irradiation with 405 nm visible light the irreversible photolytic reaction produced raseglurant. The following figure supplements are available for **Figure 1**.

The following figure supplements are available for figure 1:

**Figure supplement 1.** Supporting spectra for the synthesis of raseglurant.

**Figure supplement 2.** Supporting spectra for the synthesis of JF-NP-26.

light-dependent gain of function was demonstrated for JF-NP-26 in the mGlu$_5$-mediated IP accumulation assay. In addition, we evaluated the ability of JF-NP-26 to photomodulate the mGlu$_5$ receptor-mediated intracellular calcium accumulation in cultured cells ectopically expressing the receptor (**Figure 3B**). Agonist challenge induced a robust mGlu$_5$ receptor-mediated intracellular calcium rise both in dark and under 405 nm illumination, which was blocked by raseglurant (**Figure 3B**). Interestingly, again while JF-NP-26 was unable to restrain agonist-mediated signalling in dark conditions, it abolished mGlu$_5$ receptor-mediated intracellular calcium accumulation upon 405 nm irradiation (**Figure 3C**), thus demonstrating a light-dependent NAM activity. Next, we performed similar

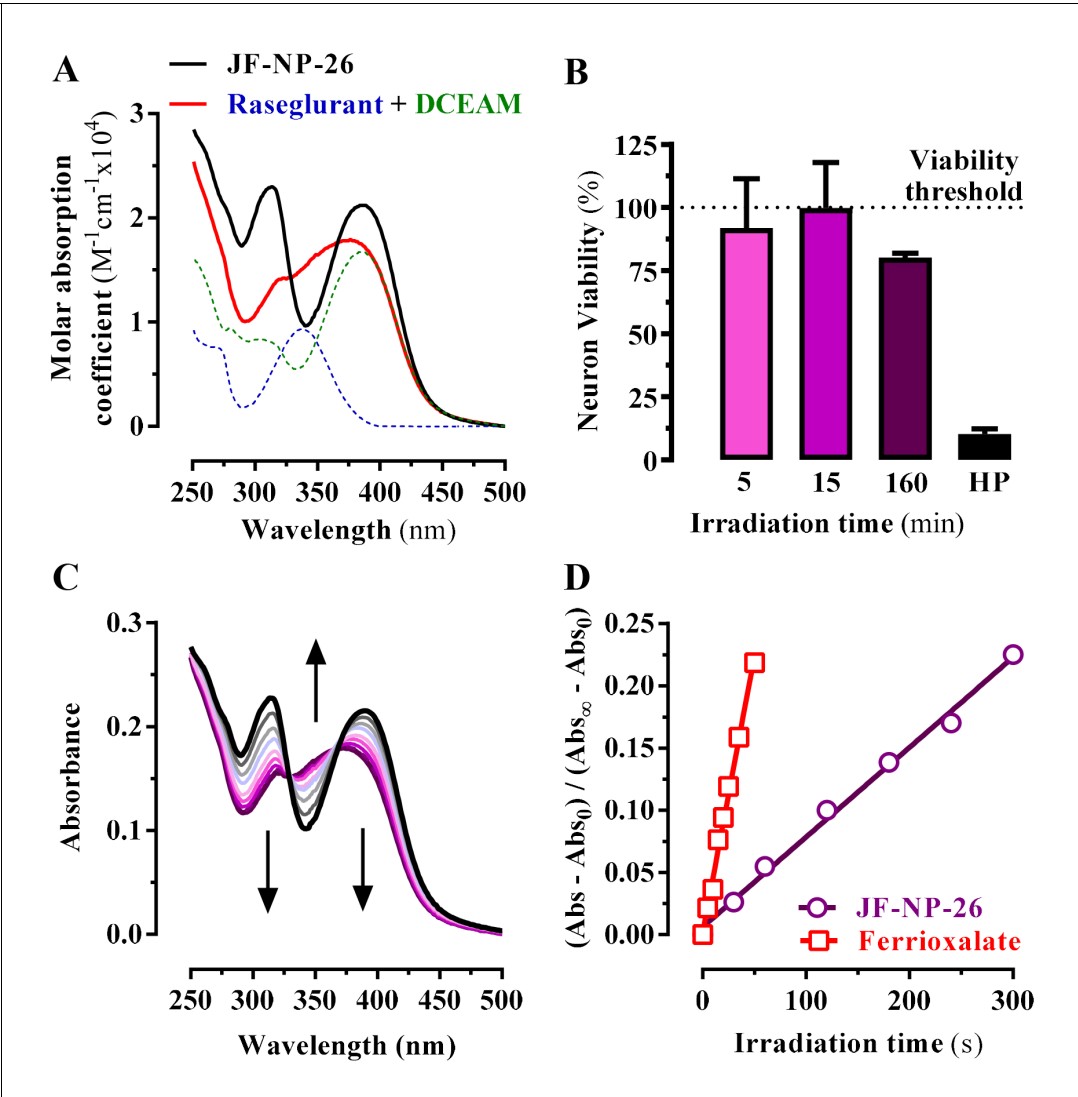

**Figure 2.** Photochemical properties of JF-NP-26. (A) UV-visible absorption spectra of raseglurant (Ras), the coumarine DEACM, and JF-NP-26 (JF) in PBS. The sum of the free Ras and DEACM spectra shows important differences relative to that of their conjugate JF (B) Neuronal viability upon 405 nm irradiation. The neuronal viability was assessed using the 3-(4,5-dimethylthiazol-2-yl)−2,5-diphenyltetrazolium bromide (MTT) assay (see Supplementary Information). Cultured neurons were irradiated with 405 nm light during 5, 15 and 160 min before the MTT incubation 3% hydrogen peroxide (HP) was used a positive control of cell death. All values were normalized using intact neurons as a 100% of viability threshold and expressed as mean ± S.E.M. of three independent experiments performed in triplicate. (C) Changes in the absorption spectrum of JF-NP-26 upon illumination in PBS with 405 nm light. The spectrum of the irradiated sample shows an excellent match with the sum of the free Ras and DEACM spectra in panel A. (D) Determination of the quantum yield of raseglurant photouncaging by comparison of the rate of this process with the rate of a standard photochemical reaction, namely ferrioxalate photoreduction ($\varphi_r$ = 1.14 at 405 nm, see Experimental Section).

calcium experiments in cultured neurons from mouse striatum (*Figure 3D*) to assess the light-dependency of NAM activity in a native system (*Figure 3E*). Results (*Figure 3F*) were comparable to those obtained in heterologous expression systems (*Figure 3C*), thus indicating that optical modulation of JF-NP-26 could also be performed in a native receptor environment.

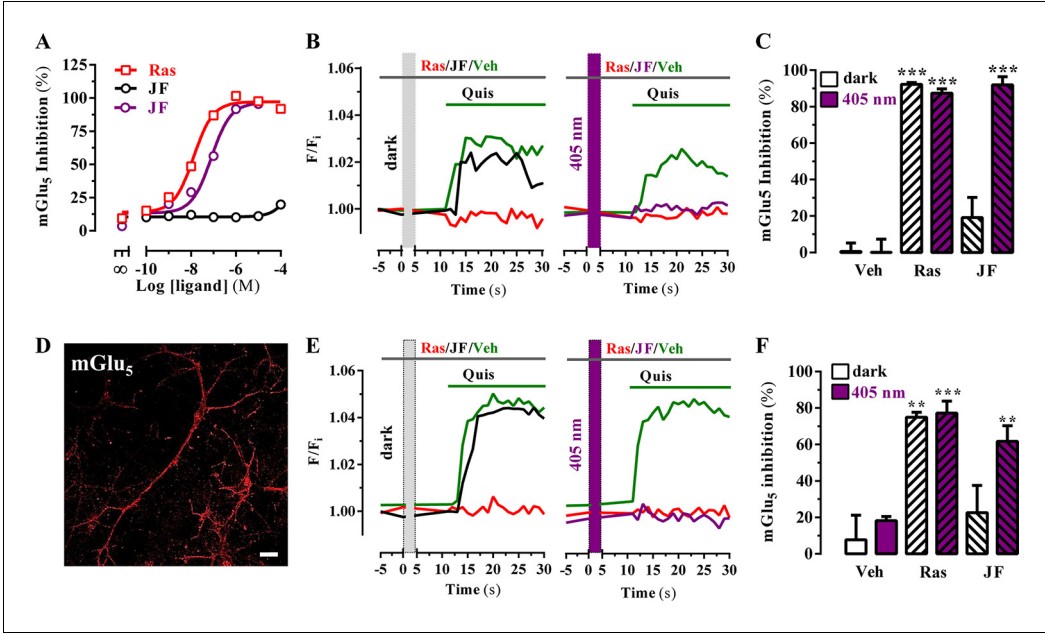

**Figure 3.** Optical modulation of mGlu5 receptor activity in cultured cells and in primary striatal neurons. (**A**) Determination of mGlu5 receptor-mediated inositol phosphate accumulation in transiently transfected HEK-293T cells. Normalized dose-response of raseglurant (Ras)- and JF-NP-26 (JF)-mediated inhibition of quisqualate (1 μM)-induced IP accumulation in dark or upon irradiation at 405 nm (violet trace) in mGlu5 receptor expressing cells. Each data point corresponds to mean± SEM of three independent experiments performed in triplicate. The pEC50 value determined for quisqualate was 7.52. (**B**) Transiently transfected HEK-293T cells with the mGlu5 receptor were pretreated with vehicle (veh), raseglurant (Ras) and JF-NP-26 (JF) at 1 μM in the dark (left panel) or upon 405 nm irradiation (right), and then challenged with quisqualate (Quis, 100 μM). (**C**) Quantification of mGlu5 receptor inhibition. The percentage of receptor inhibition is calculated as described in Materials and methods and expressed as mean ± S.E.M. (n = 10). ***p<0.001, one-way ANOVA with Dunnett's multiple comparison test using the Veh+dark condition as a control. (**D**) Expression of mGlu5 receptor in primary striatal neurons. Primary striatal neurons were fixed, permeabilized and immunostained using a rabbit anti-mGlu5 receptor (1 μg/ml) antibody. The primary antibody was detected using Cy3-conjugated donkey anti-rabbit antibody (1/200). Neurons were analyzed by single immunofluorescence with a confocal microscopy. *Scale bar*: 10 μm. (**E**) Determination of mGlu5 receptor-mediated intracellular calcium accumulation in primary striatal neurons. Neurons were incubated with vehicle (veh), raseglurant (Ras) or JF-NP-26 (JF), then kept in the dark (left panel) or irradiated at 405 nm (right panel) before being challenged with quisqualate (Quis) (see Materials and methods). (**F**) Quantification of mGlu5 receptor inhibition. The percentage of receptor inhibition is calculated as described in the Material and methods section and expressed as mean ± S.E.M. (n = 5–10). **p<0.005 and ***p<0.001, one-way ANOVA with Dunnett's multiple comparison test using the Veh+dark condition as a control.

## Peripheral and central antinociceptive action of JF-NP-26 upon in vivo photoactivation

We assessed the in vivo analgesic activity of JF-NP-26 activated by light in two established models of pain in mice: the chronic constriction injury (CCI) model of neuropathic pain and the formalin test (*Figure 4A*) (*Mogil, 2009*).

First, we proved the validity of our approach in mice subjected to unilateral CCI of the sciatic nerve, in which we explored the contribution of central mGlu5 receptors by photomodulating these receptors in the ventrobasal thalamus, a pivotal relay and processing point for somatosensory information ascending from the spinal cord to the cerebral cortex (*Kolber, 2015*; *Palazzo et al., 2014*). Accordingly, we implanted optical fibers in the brain to allow precise light delivery into the ventrobasal thalamus (*Figure 4A*). Importantly, we used a LED lamp illumination system, not requiring the use of laser sources to achieve the photorelease of the active compound, and assessed mechanical pain thresholds by means of von Frey filaments (*Figure 4A*). Interestingly, a single injection of raseglurant (10 mg/kg, i.p.) significantly increased pain thresholds in CCI mice regardless of light

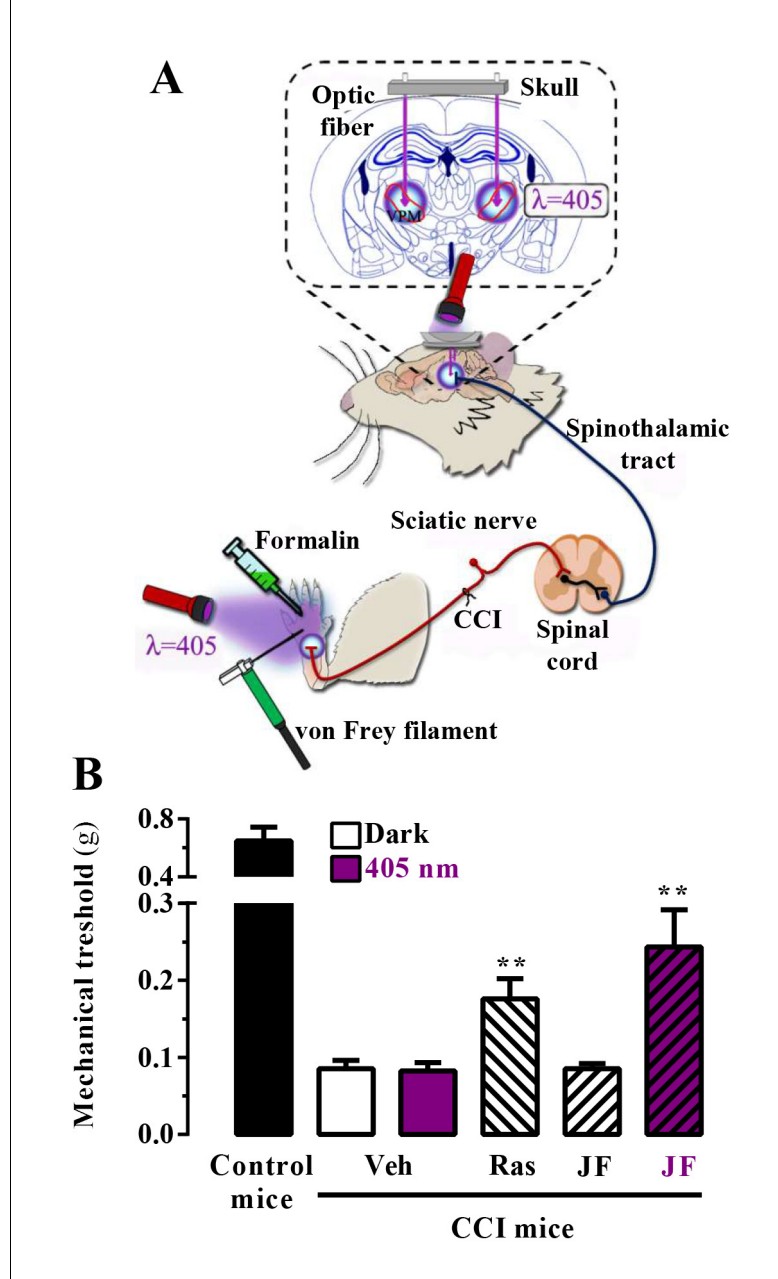

**Figure 4.** In vivo assessment of JF-NP-26 light-dependent analgesic efficacy in neuropathic pain. (**A**) Scheme showing the different LED-mediated irradiation points within the pain neuraxis (i.e. hind paw and thalamus) and pain triggering (i.e. hind paw formalin administration and chronic constriction injury -CCI- of the sciatic nerve). VPM, ventral posteromedial nucleus. (**B**) Chronic constriction injury (CCI) of the sciatic nerve causing mechanical allodynia. Mechanical thresholds were measured in 21 days post-surgery CCI mice. Thus, animals were intraperitoneally injected with vehicle (Veh, saline), raseglurant (Ras, 10 mg/kg) or JF-NP-026 (JF, 10 mg/kg) 20 min before the thalamus was irradiated at 405 nm (or dark) for 5 min. Subsequently, the mechanical thresholds were assessed before and after light irradiation. Values are means ± S.E.M. of 7–9 mice per group. **p<0.05, Student t test, JF-NP-26 dark vs. JF-NP-26 irradiated and raseglurant dark vs. Veh dark.

irradiation (*Figure 4B*; data with raseglurant after irradiation are not shown). In contrast, systemic injection of JF-NP-26 (10 mg/kg, i.p.) significantly (p<0.01) increased pain thresholds in CCI mice only after thalamic irradiation (*Figure 4B*).

On the basis of this proof of principle success, we took advantage of a different mice model of pain to further investigate the analgesic activity of light-delivered JF-NP-26 either in the periphery or in ventrobasal thalamus. The formalin test, which allows an objective analysis of pain based on nocifensive behavior, displays a first phase of pain (5 min after formalin injection in the hind paw), which models acute inflammatory pain, and a second phase (20–30 min after formalin injection), which models chronic inflammatory pain and reflects the development of central sensitization (*Mogil, 2009*). Formalin injection into the mouse hind paw induced an innate licking behavior that was significantly reduced by systemic administration of raseglurant (10 mg/kg, i.p., *Figure 5*, middle panel), thus demonstrating an antinociceptive efficacy both in phase I (70 ± 3%, p<0.001) and phase II (97 ± 1%, p<0.001) of the formalin test. We then directly irradiated the ipsilateral hind paw with external 405 nm light (*Figure 4A*) following the experimental regimen shown in *Figure 5* (upper panel). Noteworthy, while JF-NP-26 (10 mg/kg, i.p.) was unable to promote antinociception in dark conditions (*Video 1*), it elicited antinociception following direct hind paw irradiation both at phase I (54 ± 5%, p<0.001) and phase II (34 ± 5%, p<0.001) (*Figure 5*, middle panel; *Video 2*), demonstrating that JF-NP-26 was able to be peripherally photoactivated. Interestingly, under the same experimental conditions when the contralateral hind paw was irradiated no antinociceptive effect was observed (*Figure 5*, middle panel, column C), thus demonstrating that a non-site specific peripheral photoconversion of JF-NP-26 does not yield systemic raseglurant-mediated analgesia. On the other hand, we also evaluated the effects of JF-NP-26 by delivering light into the ventrobasal thalamus. As shown in *Figure 5* (lower panel), implantation of the illumination system by itself did not affect nocifensive responses in the formalin test and did not alter the analgesic activity of raseglurant, as expected. While JF-NP-26 (10 mg/kg, i.p.) was inactive under basal conditions, it was able to cause substantial analgesia both in phase I (45 ± 9%, p<0.01) and phase II (90 ± 4%, p<0.001) of the formalin test in response to thalamic irradiation (*Figure 5*, lower panel). Importantly, under the same experimental conditions when the striatum was irradiated no antinociceptive effect was observed (*Figure 5*, lower panel, column S), indicating that a CNS region of the pain neuraxis must be specifically irradiated to obtain pain control by JF-NP-26. Significantly, uncaging of JF-NP-26 in the thalamus produced a stronger analgesic effect in phase II of the formalin test, which reflects mechanisms of central sensitization. Finally, the thalamic irradiation effects of JF-NP-26 were obtained after intraperitoneal administration, demonstrating its effective brain penetration in mice. Overall, upon local and thalamic photoactivation, JF-NP-26 demonstrated analgesic effects in two different models of pain, thus raising the interesting possibility that localized light targeting of peripheral and thalamic mGlu$_5$ receptors represent a valuable strategy for the treatment of pain diseases.

## Systemic administration and in vivo photoactivation of JF-NP-26 does not impair memory in mouse

Raseglurant has been shown to possess effectiveness on migraine treatment (*Keywood and Wakefield, 2008*). However, despite good clinical efficacy, tolerability and safety in a 2-week double-blind, placebo-controlled, multicentre trial (*Zerbib et al., 2011*), the long-term administration of raseglurant in migraine patients resulted in a disappointing high incidence of hepatic transaminase abnormalities, thus halting any further clinical development (*Marin and Goadsby, 2010*). Hence, we aimed to assess if raseglurant and JF-NP-26 induced liver toxicity in mice. To this end, we determined serum levels of alanine aminotransferase (ALT) and aspartate aminotransferase (AST) after a 3-day high dose (i.e. 50 mg/kg) raseglurant and JF-NP-26 treatment, thus harsher conditions to that used in our antinociceptive experiments (i.e. single dose of 10 mg/kg, ip). Interestingly, while a 3-day acetaminophen treatment (300 mg/kg, ip) induced a significant increase in serum ALT and AST (*Figure 6A*), as described (*Zhang et al., 2015*), the 3-day raseglurant and JF-NP-26 treatment (50 mg/kg, ip) did not alter ALT and AST serum levels (*Figure 6A*). Thus, these results demonstrated that raseglurant and JF-NP-26 treatments used in our pain animal models were safe because no liver toxicity was observed.

On the other hand, it has been reported that mGlu$_5$ receptor NAMs may produce behavioural undesirable side effects. For instance, systemic administration of MPEP, a mGlu$_5$ receptor NAM, reduced reference memory in rats (*Christoffersen et al., 2008*). Thus, we aimed to explore whether

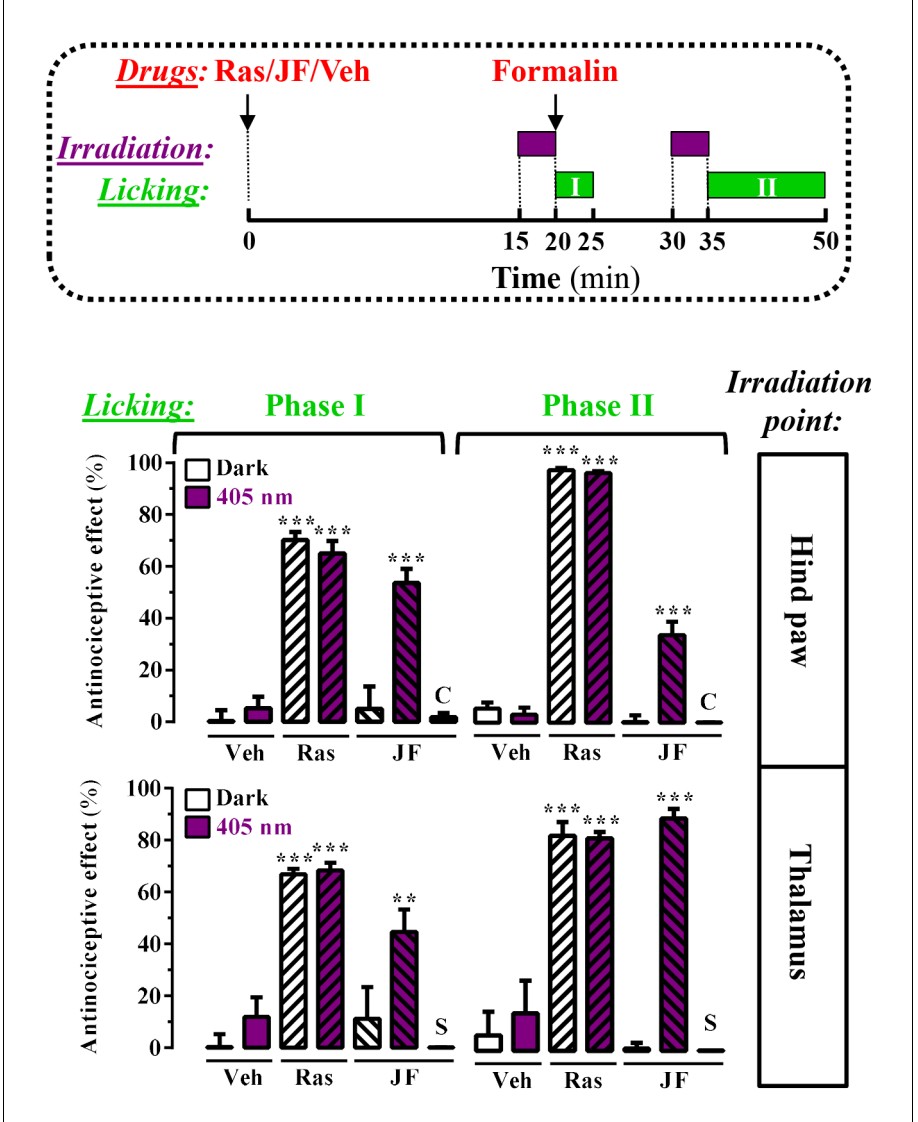

**Figure 5.** Peripheral and central light-dependent JF-NP-26-mediated antinociception in mice. In the upper panel a scheme of the irradiation regime at 405 nm light (violet rectangles) and liking recordings (green rectangles - Phase I and Phase II) in the formalin animal model of pain is shown. Thus, animals were intraperitoneally injected with vehicle (Veh, 20%DMSO + 20%tween80 in saline), raseglurant (Ras, 10 mg/kg) or JF-NP-026 (JF, 10 mg/kg) 20 min before the hind paw or the thalamus was irradiated at 405 nm (or dark) for 5 min. As a control, the contralateral hind paw (**C**) or the striatum (**S**) were also irradiated. Subsequently, the total hind paw licking was measured for 0–5 min (Phase I) and 15–30 min (Phase II) after intraplantar injection of 20 µl of formalin solution (2.5% paraformaldehide). The antinociceptive effect was calculated as the percentage of the maximum possible effect (mean ± S.E.M., n = 5–6 mice per group). **p<0.01 and ***p<0.001, one-way ANOVA with Dunnett's multiple comparison test using Veh+dark as a control.

systemic administration and local photoactivation of JF-NP-26 circumvented raseglurant-associated adverse effects. To this end, we assessed the impact of raseglurant and JF-NP-26 in mouse working/ reference memory by using the novel object recognition test. As expected, naive mice injected with vehicle spent significantly more time in exploring the novel object than the familiar object (*Figure 6B*). Interestingly, raseglurant impaired novel object recognition at the dose of 20 mg/kg (*Figure 6B*), although it was inactive at 10 mg/kg (not shown). Optic fiber implanted mice treated with vehicle or JF-NP-26 (10 or 20 mg/kg) and irradiated bilaterally at the ventrobasal thalamus (see

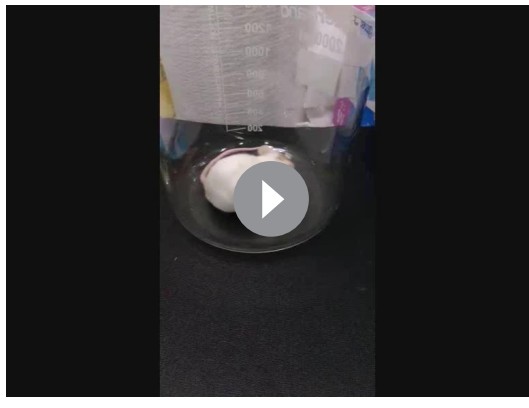

**Video 1.** Video showing formalin-induced hind paw licking in a JF-NP-26 -treated mouse upon dark conditions from *Figure 5*.　　Mouse, in a glass beaker (15 cm ø x 20 cm height), was administered with JF-NP-26 (10 mg/kg, i.p.). After 20 min, the LED-based irradiation system was attached to the thalamic implanted optic fiber and mock irradiated (dark) during 5 min. Then, formalin solution (20 µL) was intraplantarly (i.pl.) injected and total licking or biting time of the hind paw was recorded. A 30 s' video fragment exemplifying the formalin test is shown.

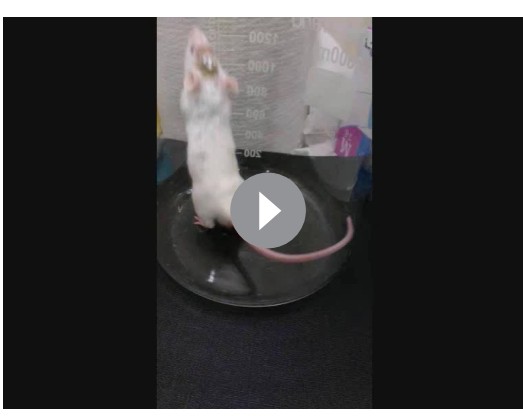

**Video 2.** Video showing formalin-induced hind paw licking in a JF-NP-26-treated mouse upon 405 nm irradiation from *Figure 5*.　　Mouse, in a glass beaker (15 cm ø x 20 cm height), was administered with JF-NP-26 (10 mg/kg, i.p.). After 20 min, the LED-based irradiation system was attached to the thalamic implanted optic fiber and irradiated with 405 nm visible light during 5 min. Then, formalin solution (20 µL) was intraplantarly (i.pl.) injected and total licking or biting time of the hind paw was recorded. A 30 s' video fragment exemplifying the formalin test is shown.

above) showed a normal performance in the novel object recognition test (*Figure 6A*). Overall, these results demonstrated that neither the systemic administration nor the thalamic photoactivation of JF-NP-26 cause memory impairment in mice.

## Discussion

We report a new pharmacological approach based on analgesic ligands endowed with light-dependent activity. This strategy allows the photocontrol of mGlu$_5$ receptors at different stations of the pain neuraxis.

To our knowledge, JF-NP-26 is the first caged mGlu$_5$ receptor NAM that has been shown to induce light-dependent analgesia in models of inflammatory and neuropathic pain in freely behaving animals. The ability to resolve analgesia with a fine resolution in time and space confers to our experimental approach a high significance for further development of new caged drugs acting at different levels of the pain neuraxis.

Raseglurant is a mGlu$_5$ receptor NAM that has been shown to possess efficacy on the treatment of migraine (*Keywood and Wakefield, 2008*). However, its clinical development was discontinued because of liver toxicity (*Marin and Goadsby, 2010*). Accordingly, we synthesized a caged compound (JF-NP-26) that upon illumination would release raseglurant only in the areas of interest, thus avoiding potential systemic undesired effects associated to mGlu$_5$ receptor functioning (i.e. memory impairment). Of note, uncaging of JF-NP-26 was designed to be elicited with light pulses in the visible spectrum (405 nm). This property differentiates JF-NP-26 from the majority of caged compounds that require UV light for activation. This is highly advantageous for translational research because light within the visible spectrum has a good tissue penetration and causes less tissue damage with respect to UV light (*Regan and Parrish, 1993*). Remarkably our LED-based illumination allowed the effective transdermal uncaging of JF-NP-26 at the mouse hind paw, just where many noxious stimuli are detected and transduced by peripheral nociceptors. In fact, mGlu$_5$ receptors are expressed in peripheral nociceptors, where they contribute to mechanisms of nociceptive sensitization by enhancing the activity of TRPV1 channels through a complex mechanism mediated by inositol phospholipid hydrolysis, prostaglandin formation, and protein kinase A-dependent phosphorylation of TRPV1 (*Bhave et al., 2001*; *Karim et al., 2001*; *Neugebauer, 2001*; *Hu et al., 2002*). Thus, raseglurant

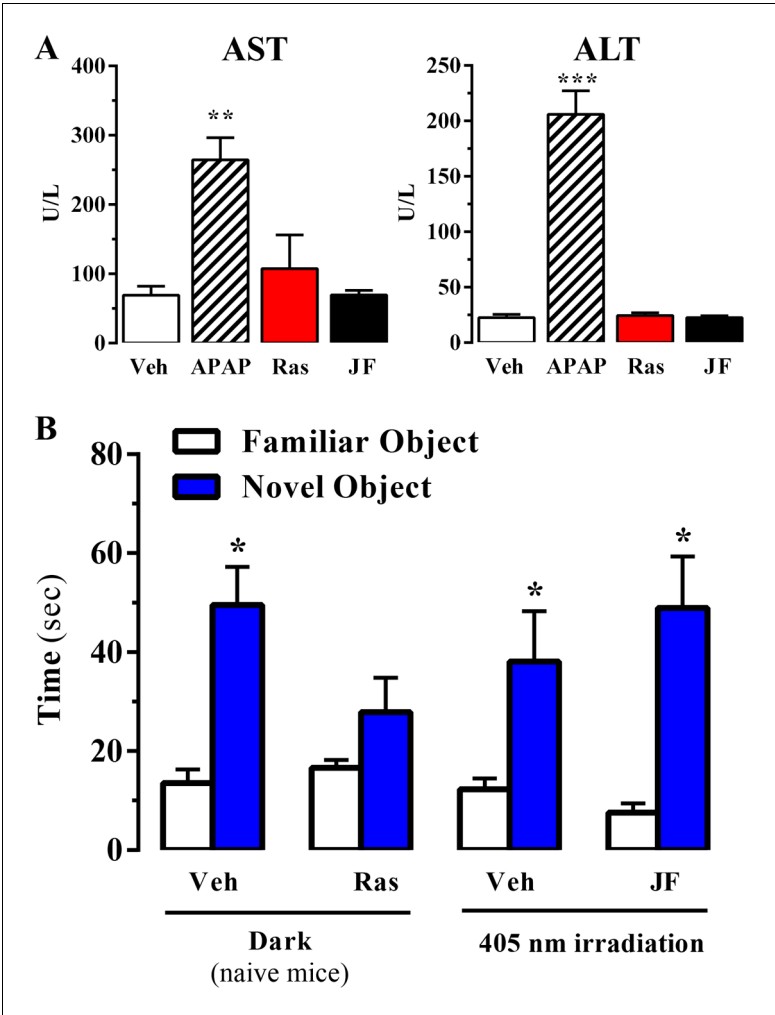

**Figure 6.** Effect of systemic administration and in vivo photoactivation of JF-NP-26 in liver toxicity and memory in mouse. (A) Effect of Raseglurant and JF-NP-26 on serum transaminases. Mice were treated with vehicle (veh), acetaminophen treatment (APAP; 300 mg/kg, ip), raseglurant (Ras; 50 mg/kg, ip) or JF-NP-26 (JF; 50 mg/kg, ip) for three consecutive days. Serum levels of alanine aminotransferase (ALT) and aspartate aminotransferase (AST) were determined as described in Materials and methods. The results are expressed as means ± S.E.M. of three mice per group. **p<0.01, ***p<0.001 (One-Way ANOVA followed by Dunnett's post hoc test using the vehicle condition as a control. (B) Novel object recognition test. The time spent in exploring novel and familiar objects were measured 35 min after drug administration. Naive mice were intraperitoneally injected with vehicle (6% DMSO +6% tween80 in saline) or raseglurant (10 or 20 mg/kg). Mice bilaterally implanted with optic fibers in the thalamus were injected with vehicle or JF-NP-26 3 (10 or 20 mg/kg) 15 min before light irradiatiation (405 nm for 5 min). Values are means ± S.E.M. of 4–7 mice per group. *p<0.05 (One-Way ANOVA + Fisher's LSD test) *vs.* the respective values of the time spent with the familiar object ($F_{(7,38)}$ = 7.28). Doses of 10 mg/kg of either raseglurant in naive mice or JF-NP-26 three in light-irradiated mice did not affect recognition memory. These data are not included in the graph.

released from JF-NP-26 in response to light irradiation in the hind paw may interact with $mGlu_5$ receptors present in peripheral nociceptors, thereby causing analgesia in the formalin test. In line with this, intraplantar JF-NP-26 uncaging showed higher antinociceptive efficacy in the first phase of the formalin test, which reflects the activation of peripheral nociceptors (*Hunskaar and Hole, 1987*).

One of the major findings of the present work consists of the local photoactivation of JF-NP-26 to give the active compound after its systemic (i.p.) administration. Accordingly, mice were systemically injected with JF-NP-26, and fast antinociceptive effects were produced by light irradiation in the ventrobasal thalamus, the main relay station of the ascending pain pathway modulating the

sensory, cognitive, and emotional components of pain. Light-mediated delivery of raseglurant in the thalamus showed a greater antinociceptive efficacy during the second phase of the formalin test, which reflects the development of central sensitization to inflammatory pain. Hence, our data suggested that mGlu5 receptors located at two different stations of the pain neuraxis (i.e., peripheral nociceptors and ventrobasal thalamic nuclei) might be contributing differentially to acute inflammatory pain and mechanisms of central sensitization Importantly, JF-NP-26 also showed fast analgesic activity in the CCI model of neuropathic pain in response to thalamic irradiation. Neuropathic pain is characterized by a robust and long-lasting nociceptive sensitization that reflects a maladaptive process of activity-dependent synaptic plasticity (*Basbaum et al., 2009*). Interestingly, mGlu5 receptors participate on the expression of this pathological form of synaptic plasticity at different levels of the pain neuraxis, from the spinal cord to the regions of the pain matrix including the amygdala and cerebral cortex (*Neugebauer et al., 2003*; *Li and Neugebauer, 2004*; *Hu et al., 2007*; *Hu and Gereau, 2011*; *Lin et al., 2015*). Our data indicate that thalamic mGlu5 receptors are critical for the expression of nociceptive sensitization and can be specifically targeted by analgesic drugs in the treatment of neuropathic pain. Overall, our research provided valuable information regarding the contribution of mGlu5 receptors to the processing of nociceptive transmission at specific anatomic loci within the pain neuraxis (e.g., central vs. peripheral) as well as its participation in the modulation of both acute inflammatory and chronic neuropathic pain.

In conclusion, systemic administration of inactive drugs that are activated by light on demand may provide a powerful strategy to reach a rapid control of pain limiting the development of dose- and mechanism-related adverse effects that are typically encountered with conventional analgesic drugs. Our positive results constitute a proof of principle that mGlu5 receptor NAM uncaging may be a suitable therapeutic approach for the local control of pain, which could be applied in general to conditions of severe pain that are difficult to treat, and, in particular, to conditions of fast-onset severe pain that requires a rapid and highly localized treatment. However, the transition from pre- to clinical settings still has to solve a number of practical problems, including assessment of photoactive compound stability and toxicity, as well as the development of implantable and remotely controlled LEDs covering the photochemical needs to release the active compounds with light.

## Materials and methods

### Synthesis of drugs

All the chemicals and solvents were provided from commercial suppliers and used without purification, except the anhydrous solvents, which were treated previously through a system of solvent purification (*PureSolv*), degasified with inert gases and dried over alumina or molecular sieve (dimethyl formamide). Dry triethylamine was distilled over calcium hydride. Building blocks 2-bromo-4,6-dimethylpyridin-3-amine (OR7028) and 1-ethynyl-3-fluorobenzene (PC5956, 98%), were purchased from *Apollo Scientific*, bis(triphenylphosphine)palladium (II) dichloride (412740, $\geq$ 99% trace metal basis) and copper iodide (03140, purum, $\geq$99,5%) from *Sigma-Aldrich* (St. Louis, MO, USA). DCEAM coumarin was obtained following the protocol reported in the literature[1], starting from 7-diethylamino-4-methylcoumarin (M063, $\geq$98%), purchased from TCI. Reactions were monitored by thin layer chromatography (60 F, 0.2 mm, *Macherey-Nagel*) by visualisation under 254 and/or 365 nm lamp. *Reverse phase* column chromatography KP-C18-HS 12g (*Biotage*), automated with *Isolera One* with UV-Vis detection (*Biotage*) using CV (Column volumes) as units for elution gradient, corresponding 1 CV to 1,28 min at 14 mL/min of flux. *Flash* column chromatography SNAP KP Sil 50 microns (*Biotage*), automated with *IsoleraOne* with UV-Vis detection (*Biotage*). mp: melting point B-545 (*Büchi*), ramp 2 °C/min. NMR: *Variant-Mercury 400* MHz (*Agilent Technologies*). Chemical shifts are reported in parts per million (ppm) against the reference compound tetramethysilane using the signal of the residual non-deuterated solvent (Chloroform $\delta$ = 7.26 ppm ($^1$H), $\delta$ = 77.16 ppm ($^{13}$C), DMSO $\delta$ = 2.50 ppm ($^1$H) Methanol $\delta$ = 3.34 ppm ($^1$H), $\delta$ = 49 ppm ($^{13}$C)). High-Performance Liquid Chromatography (HPLC) *Alliance 2695* separation module (*Waters*) coupled to *Waters 2996* photodiode detector (DAD) (*Waters*), 5 µL of sample 2.5 mM in DMSO was injected, using a *Kinetex* C18 2.6 µm 4.6 $\times$ 50 mm (*Phenomenex*) column. The mobile phase used was a mixture of A = $NaH_2PO_4$/$Na_2HPO_4$ aqueous buffer 10 mM pH = 7 and B = methanol, with the method described as follows: flow 1 mL/min 5%B-70%B 1 min, 70%B-80%B 1 min, 80%B-100%B 6 min, 100%B 1 min, runtime 15

min. Ultrahigh-Performance Liquid Chromatography (UPLC) *Aquity(Waters)* coupled to an *Acquity TUV* UV-Vis detector (*Waters*) and a *LCT Premier Orthogonal Accelerated Time of Flight Mass Spectrometer* (TOF) (*Waters*). HRMS: UPLC/MS chromatograms were obtained by injection of 2 'L of sample 25 'M in AcN/DMSO 99:1, using a*Acquity UPLC BEH* C18 1.7 'm 2.1 × 100 mm column (*Waters*). The mobile phase used was a mixture of A = aqueous formic acid 20 mM and B = formic acid 20 nM in acetonitrile, with the method described as follows: flow 0.3 mL/min 5–95%B 2.69 min, 95%B 6 min, runtime 14 min. Data from mass spectra were analyzed by electrospray ionization in positive and negative mode. Spectra were scanned between 50 and 1500 Da with values every 0.2 s and peaks are given m/z (% of basis peak). ESI were analyzed by FIA (flux injected analysis) with *Acquity* (*Waters*) coupled to LCT Premier Orthogonal Accelerated Time of Flight Mass Spectrometer (TOF) (*Waters*). Data were obtained in the same conditions as for samples of UPLC/MS.

The synthesis of 2-((3-fluorophenyl)ethynyl)−4,6-dimethylpyridin-3-amine, raseglurant (ADX-10059) (*Figure 1*) was previously described in the literature (*Bolea et al., 2004*) using a general conditions of Sonogashira reaction. A solution of 2-bromo-4,6-dimethylpyridin-3-amine (200 mg, 0.995 mmol), bis(triphenylphosphine)palladium (II) dichloride (35 mg,0,05 mmol) and copper iodide (10 mg, 0,05 mmol), previously purged with argon, in 1 mL of anhydrous DMF, 1-ethynyl-3-fluorobenzene (0,12 mL,1.094 mmol) and dry triethylamine (0,42 mL, 2.98 mmol) were added, and the reaction mixture was stirred at 40°C for 8 hr. After this time, 40 mL of ethyl acetate was added, and the mixture was washed with 40 mL of saturated solution of $NaHCO_3$ and 40 mL of brine, the organic layer was dried over $Na_2SO_4$ filtered and evaporated under vacuum. The residue was purified trough flash column chromatography with ethyl acetate-hexane 1:4. A palid brown solid was isolated (197 mg, 83%). A portion of this compound was dissolved in ether and HCl 1N was added, the precipitate was collected by filtration, to give the hydrochloride salt as a yellow solid. $^1$H NMR (400 MHz, DMSO-d6) δ 7.75–7.71 (m, 1H), 7.63 (dd, J = 7.6, 1.2 Hz, 1H), 7.56 (td, J = 8.0, 5.9 Hz, 1H), 7.50 (s, 1H), 7.39 (td, J = 8.6, 2.5 Hz, 1H), 4.00 (s, 4H), 2.54 (s, 3H), 2.34 (s, 3H). 13C NMR (101 MHz, DMSO-d6) δ 162.92, 160.49, 145.61, 141.44, 140.80, 131.08, 131.00, 128.21, 128.18, 127.78, 122.66, 122.56, 118.59, 118.36, 117.57, 117.36, 113.15, 101.11, 79.92, 18.37, 17.88. HRMS (m/z): $[M+H]^+$-calcd. for $C_{15}H_{13}FN_2$, 241.1141; found, 241.1112 HPLC/DAD: purity (abs = 254 nm)=100%; RT = 3.34 min.

The synthesis of synthesis of (7-(diethylamino)-2-oxo-2H-chromen-4-yl)methyl (2-((3-fluorophenyl) ethynyl)-4,6-dimethylpyridin-3-yl)carbamate, JF-NP-26 (*Figure 1*), was performed using conventional chemical methods. A solution of triphosgene (100 mg, 0.336 mmol) in 2 mL of toluene, a mixture of 2-((3-fluorophenyl)ethynyl)-4,6-dimethylpyridin-3-amine (218 mg, 0.90 mmol) and dry triethylamine (0,22 mL, 1.68 mmol) in 2 mL of toluene was added, and the reaction mixture was stirred over 3 hr (the reaction was monitored by TLC).

Afterwards, the reaction mixture was purged with a gentle flow of nitrogen over 20 min to remove the phosgene produced during the reaction, the crude residue was dissolved in 2 mL of toluene and a solution of 7-(diethylamino)-4-(hydroxymethyl)-2H-chromen-2-one (83 mg, 0.337 mmol) and sodium hydride (14 mg, 0.337 mmol) in 1 mL of THF previously stirred at room temperature for 10 min, was added and heated at 100°C overnight. The mixture was evaporated under vacuum and 40 mL of ethyl acetate were added, washed with 40 mL of brine, dried over $Na_2SO_4$, filtered and evaporated under vacuum. The residue was purified by flash column chromatography in ethyl acetate-hexane 3:2. A yellow-orange solid was obtained and it was purified again at isolerabiotage in reverse phase conditions: 4VC ($H_2O$:Acetonitrile 95:5), 3VC ($H_2O$:Acetonitrile 40:60), 10 VC ($H_2O$: Acetonitrile 0:100), the crude was lyophilized overnight. A yellow solid was obtained (95 mg, 55%). Mp: 77–79°C,1H NMR (400 MHz, Methanol-d4) δ 7.50 (s, 1H), 7.41–7.29 (m, 3H), 7.28–7.21 (m, 2H), 7.19–7.12 (m, 1H), 6.67 (d, J = 9.1 Hz, 1H), 6.53 (s, 1H), 6.19 (s, 1H), 5.42 (s, 2H), 3.46 (q, J = 7.1 Hz, 6H), 1.20 (t,J = 7.1 Hz,9H). 13C NMR (101 MHz, cd3od) δ 163.55, 156.93, 156.10, 151.01, 147.16, 130.23, 130.14, 127.60, 127.57, 125.33, 124.78, 118.02, 117.78, 116.28, 116.07, 108.94, 105.74, 96.83, 44.15, 21.87, 16.38, 11.28. HRMS (m/z): $[M+H]^+$calcd. for $C_{30}H_{28}FN_3O_4$, 514.2142; found, 514.2125 HPLC/DAD: purity (abs = 254 nm)=100%; RT = 5.88 min.

## Photochemical characterization

The absorption spectra of raseglurant and JF-NP-026 were recorded in DMSO solutions (100 μM) with a Varian Cary 300 UV-Vis spectrophotometer (Agilent Technologies). Full absorption spectra were obtained by scanning between 300 nm to 600 nm with an average time of 33 ms at 5 nm

intervals. To determine the optimal illumination wavelength for uncaging, we recorded UV-Vis spectra plotting the values giving a maximum absorption at 385 nm for JF-NP-026. The photouncaging quantum yield ($\varphi_{chem}$) was measured by comparison of the rate of photouncaging ($k_{chem}$) with the rate of potassium ferrioxalate photoreduction ($k_r$) upon excitation at 405 nm. The solutions of JF-NP-26 and potassium ferrioxalate were optically-matched at 405 nm and the quantum yield of potassium ferrioxalate reduction was taken as $\varphi_r$ = 1.14 (see [*Kuhn et al., 1989*]). Potassium ferrioxalate was synthesized as described elsewhere (*Klán et al., 2013*).

## Cell culture

Human embryonic kindney (HEK) 293 T cells obtained from ATCC (CRL-321, RRID:CVCL_0063) were transiently transfected with the mGlu$_5$ receptor either by polyethylenimine (PEI) method or by electroporation technique. Then, 24 hr after transfection cells were seeded in a 96-well plate (10.000 cells/well) using Dulbecco's modified eagles medium (Sigma-Aldrich) supplemented with 100 U/mL penicillin (Biowest), 100 µg/mL streptomycin (Biowest), 10% v/v fetal bovine serum (Invitrogen), 1 mM pyruvic acid (Biowest), non-essential aminoacids (Biowest) and 2 mM L-Glutamine (Biowest). After 24 hr, intracellular calcium determinations experiments were performed. Cells were kept at 5% $CO_2$, 37°C, and 95% humidity conditions. Cells were tested for mycoplasma content, thus only mycoplasma-free cells were used.

## Striatal primary neurons and immunocytochemistry

Primary neurons from striatum were cultured from E18 CD-1 mice embryos as previously described (*Gratacos et al., 2001*). In brief, the dissected striatum was treated with 1.25% trypsin (Sigma-Aldrich) for 10 min before being mechanically dissociated with a flame polished Pasteur pipette. Neurons were plated onto poly-D-lysine (0.1 mg/mL)/laminin (0.01 mg/ml)-coated 96-well black plate at a density of 80,000 cells/cm$^2$in minimum essential medium (Invitrogen) supplemented with 10% horse serum(Invitrogen), 10% bovine serum(Invitrogen), 1 mM pyruvic acid (Biowest), and 0.59% glucose (Biowest). After 4–14 hr, the medium was substituted with Neurobasal medium (Invitrogen) supplemented with penicillin (100 U/mL) (Biowest), streptomycin (100 µg/mL) (Biowest), 0.59% glucose (Biowest), and B27 supplement (Invitrogen). Neurons were kept at 5% $CO_2$, 37°C and 95% humidity for 21 days before the calcium determination experiments.

For immunocytochemistry, primary neurons from striatum growing on coverslips were fixed in 4% paraformaldehyde for 15 min and exposed to a rabbit anti-mGlu$_5$ receptor antibody (1 µg/ml; Millipore, Billerica, MA, USA; RRID:AB_2295173). Primary antibodies were detected using a Cy3-conjugated donkey anti-rabbit antibody (1/200; Jackson ImmunoResearch Laboratories Inc., West Grove, PA, USA). Coverslips were rinsed for 30 min, mounted with Vectashield immunofluorescence medium (Vector Laboratories, Peterborough, UK) and examined using a Leica TCS 4D confocal scanning laser microscope (Leica Lasertechnik GmbH, Heidelberg, Germany) (*Luján et al., 2001*).

## Neuronal viability

The neuronal viability upon irradiation with 405 nm was calculated using the 3-(4,5-dimethylthiazol-2-yl)$-$2,5-diphenyltetrazolium bromide (MTT) assay (adapted from [*Lopachev et al., 2016*]). In brief, cultured neurons in 96 well plates were irradiated with 405 nm light for different periods of time (5, 15 or 160 min) before the incubation with 0.5 mg/mL of MTT. Subsequently, the medium was removed and 100 µL of dimethylsulfoxide (DMSO) was added to each well. The absorbance at 540 nm wavelength of each sample was measured in a POLARstar Omega multi-mode microplate reader (BMG Labtech). All values were normalized using intact neurons as a 100% of viability threshold. 3% hydrogen peroxide (HP) was used a positive control of cell death.

## Inositol phosphate determinations

The mGlu$_5$-mediated inositol phosphate (IP) accumulation was determined using the FRET-based assay IP-One HTRF kit (Cisbio Bioassays) according to the manufacturer's instructions (*Trinquet et al., 2006*). Glutamate levels were maintained at minimal concentrations by co-expressing the excitatory amino acid transporter one (EAAC1). Importantly, all the mGlu$_5$ receptor contained a haemagglutinin (HA) epitope tag in their N terminus to allow cell surface expression by ELISA for subsequent normalization. After transfection cells were seeded in black clear-bottom 96-

well plates at a concentration of $1.5 \times 10^5$ cells/well. The medium was replaced by glutamate-free DMEM GlutaMAX-I (Invitrogen) after 6 hr. Next day cells were challenged with increasing concentrations of the test compounds in the presence of quisqualate $EC_{80}$ (1 μM) for 30 min, at 37°C and 5% $CO_2$ before we determined IP accumulation (*Trinquet et al., 2006*). When necessary the compounds were irradiated at 405 nm for 5 min before being added to the cells. Finally, the cells were washed once with stimulation buffer (Cisbio Bioassays) to remove any interfering chromophore and then the cells were lysed for IP determination. The IP associated fluorescence was determined in a RUBYstar multimode microplate reader (BMG Labtech).

## Intracellular calcium determinations

Fluo-4 NW Calcium Assay Kit (ThermoFisher Scientific) was used to measure intracellular calcium accumulation. In brief, HEK-293T cells or striatal primary neurons grown in 96-well black plates were incubated with Fluo4 NW as indicated by the manufacturer. Subsequently, cells were pretreated for 5 min with vehicle (HBSS), raseglurant or JF-NP-26 (1 μM) and then maintained in the dark or irradiated at 405 nm for 5 min before being challenged with a saturating concentration (100 μM) of quisqualate. Real-time Fluo4 NW emission (i.e. 535 nm) was measured after 485 nm excitation in a POLARstar Omega multi-mode microplate reader.

The results were expressed as the percentage of mGlu$_5$receptor inhibition induced by the treatments following the equation:

$$\text{Inhibition}(\%) = [(\text{AUC}^{\text{veh}} - \text{AUC}^{\text{drug}})/\text{AUC}^{\text{veh}}] \times 100$$

Where $\text{AUC}^{\text{veh}}$ and $\text{AUC}^{\text{drug}}$ represent the area under curve value in the vehicle- and drug-treated conditions, respectively.

## Animals

Animals were housed and tested in compliance with the guidelines described in the Guide for the Care and Use of Laboratory Animals (*Clark et al., 1996*) and following the European Union directives (2010/63/EU). All efforts were made to minimize animal suffering and the number of animals used. For the formalin animal model of pain adult male CD-1 mice (Charles River Laboratories, L'Arbresle, France; RRID:MGI:3785721) weighing 20–25 g were used and for the neuropathic pain animal model adult male C57BL/6J mice (Charles River, Calco, Italy; RRID:IMSR_JAX:000664) weighing 20–25 g were employed. All animals were housed in groups of five in standard cages with ad-libitum access to food and water and maintained under 12 hr dark/light cycle (starting at 7:30 AM), 22°C temperature, and 66% humidity (standard conditions). All animal experimentation was carried out by a researcher blind to drug treatments.

## Brain optic fiber implantation

Mice were anesthetized with a combination of ketamine/xylacine (dose of 100 mg/kg body weight for ketamine and 10 mg/kg for xylacine) and implanted with optic fibers (TFC_400/475–0.53_3.5 mm_TSM4.0_B45, Doric Lenses Inc., Quebec, Canada) using dental cement and chirurgic screws (Agnthos, Lidingö, Sweden) in a Kopf or Stoelting stereotaxic frame (Stoelting Co., Wood Dale, IL, USA). The site of implantation was either the left and right thalamus (coordinates: −1.8 mm posterior to the bregma, ±1.5 mm lateral to the midline, 3.5 mm ventral from the surface of skull) or the left and right striatum (coordinates: +1.2 X mm posterior to the bregma, ±1.5 mm lateral to the midline, 3.5 mm ventral from the surface of skull), following to the atlas of Paxinos and Franklin (*Franklin and Paxinos, 2008*). After optic fiber implantation mice were treated with Metacam (BoehringerIngelheim, Ingelheim am Rhein, Germany) during three days and they were checked routinely for one week.

## Measurement of liver function

Serum ALT and AST activities were determined by using an Olympus AU400 Automatic Biochemical Analyzer (Olympus, Tokyo, Japan) in the Clinical Veterinary Service at the Autonomous University of Barcelona.

## Formalin animal model of pain

The formalin animal model of pain was performed as previously described (*Lapa et al., 2009*) and explained in more detail at Bio-protocol (*López-Cano et al., 2017*). In brief, mice (n = 5–6) were administered intraperitoneally (i.p.) with vehicle, raseglurant or JF-NP-26 20 min before a diluted formalin solution (20 µL of 2.5% formalin/0.92% formaldehyde, Sigma-Aldrich) was intraplantarly (i.pl.) injected in the mid-plantar surface of the right hind paw of the mouse. The formalin-induced nociceptive behaviour in dark and light conditions was quantified as the time spent licking or biting the injected paw during the 30 min after the injection of formalin. The initial acute phase (0–5 min; phase I), reflecting the acute peripheral pain, was followed by a relatively short quiescent period, which was then followed by a prolonged response (15–30 min; phase II) which is related to the development of central nociceptive sensitization. For the peripheral JF-NP-026 uncaging the hind paw was directly irradiated with a LED-based fiberoptic system (Doric Lenses Inc.) at 405 nm light (or dark) for 5 min before the recording of each phase. For the central nervous system uncaging both thalamic nuclei were irradiated through the brain implanted optic fibers (Doric Lenses Inc.) with 405 nm light (or dark) for 5 min before the recording of each phase. The 405 nm light was administered at 2000 mA intensity and 500 Hz frequency.

The antinociception induced by the treatments in the formalin test was calculated with the equation:

$$\mathrm{Antinociceptive\ effect}(\%) = [(\mathrm{LTV} - \mathrm{LTD})/\mathrm{LTV}] \times 100$$

where $\mathrm{LTV}$ and $\mathrm{LTD}$ represent the licking/biting time in the vehicle- and drug-treated animals, respectively.

## Mechanical allodynia assessment in a neuropathic pain animal model

The assessment of the mechanical allodynia in the chronic constriction injury (CCI) of the sciatic nerve model was explained in more detail at Bio-protocol (*Notartomaso et al., 2018*). In brief, the sciatic nerve CCI induction was carried out under isoflurane anesthesia (5% for induction and 2% for maintenance), using a modified version of a previously described method (*Bennett and Xie, 1988*). The biceps femoris and the gluteus superficialis were separated by blunt dissection, and the right sciatic nerve was exposed. CCI was produced by tying one ligature (6–0 silk, Ethicon, LLC, San Lorenzo, PR, USA) around the sciatic nerve. The ligature was tied loosely around the nerve, until it elicited a brief twitch in the respective hind limb, which prevented over-tightening of the ligation, taking care to preserve epineural circulation. The incision was cleaned and the skin was closed with 2–3 ligatures of 5–0 dermalon. After performing the ligature of the sciatic nerve, during the same anesthesia session, mice were implanted with optic fibers (see above). After surgery, mice returned to their home cages and checked routinely for 72 hr.

Mechanical allodynia was assessed 7, 14 and 21 days after surgery by measuring the hind paw withdrawal response to von Frey filament stimulation. Mice were placed in a dark box (20 x 20 × 40 cm) with a wire grid bottom through which the von Frey filaments (North Coast Medical, Inc., San Jose, CA, USA), bending force range from 0.008 to 3.5 g, were applied by using a modified version of the up-down paradigm previously described (*Chaplan et al., 1994*). Briefly, lack of response to a filament indicated the next higher filament in the following stimulation, whereas a positive response indicated the next lower filament. Each filament was applied and pressed perpendicularly to the plantar surface of the hind paw until it bent for five times with a 3 min interval. The filament that evoked at least three paw withdrawals was assigned as the pain threshold in grams. Mice were treated either with raseglurant (10 mg/kg, i.p. in 6% DMSO, 6% Tween 80 in saline) orJF-NP-26, (10 mg/kg, i.p. in 6% DMSO, 6% Tween 80 in saline). Both thalamic nucleus were irradiated (or dark) through the brain implanted optic fibers (Doric Lenses Inc.) with 405 nm light (or dark) for 5 min, 20 min after drug administration. The light was administered at 2000 mA intensity and 500 Hz frequency and the mechanical thresholds were quantified before and after light irradiation.

## Novel object recognition test

To evaluate reference memory, we performed the novel object recognition test as previously described (*Christoffersen et al., 2008*) with minor modifications. In brief, mice were divided into six subgroups: vehicle-injected mice, raseglurant (10 or 20 mg/kg, i.p.)-injected mice, vehicle-injected

mice implanted with optic fibers and JF-NP-26 (10 or 20 mg/kg, i.p.)-injected mice implanted with optic fibers. All mice were injected 20 min before the training session. On the first three days, all mice were handled for 3 min once daily and 24 hr later they were habituated for 20 min to the empty arena. On day 5, mice underwent a 5 min training period. Afterwards, mice return to their home cages for 5 min, and were subsequently subjected to a 5 min retention test. The 405 nm light was administered at 2000 mA intensity and 500 Hz frequency in vehicle and JF-NP-26-injected mice 5 min immediately before the training session. Two identical objects were used as familiar objects during the training trial and the third object was used as novel object during the 5 min retention session. As an index of recognition memory, the time spent exploring the familiar and novel object was measured during the retention session.

## Statistics

The number of samples (n) in each experimental condition is indicated in figure legends. When two experimental conditions were compared, statistical analysis was performed using an unpaired $t$ test. Otherwise, statistical analysis was performed by one-way analysis of variance (ANOVA) followed by Bonferroni post-hoc test. In cell experiments and formalin test results Dunnett's multiple comparison post-hoc test was performed using Veh+dark as a control. Statistical significance was set as $p < 0.05$.

## Acknowledgements

This work was supported by MINECO/ISCIII (SAF2014-55700-P, PCIN-2013–019 C03-03 and PIE14/00034), the Catalan Government (2014 SGR 1054), ICREA (ICREA Academia-2010), Fundació la Marató de TV3 (Grant 20152031) and IWT (SBO-140028) to FC. MINECO (PCIN-2013–018 C03-02 and SAF2014-58396-R) to JG. MINECO (PCIN-2013–017 C03-01 and CTQ2014-57020-R), the Catalan Government (2014SGR109 and 2014CTP0002) to AL. ERANET Neuron project 'LIGHTPAIN' (to AL, JG, PC and FN). MINECO (CTQ2013-48767-C3-1-R and CTQ2015-71896-REDT), the Catalan Government (2014SGR304) to SN RB-O thanks the European Social Funds and the SUR del DEC de la Generalitat de Catalunya for a predoctoral fellowship (2016 FI_B1 00021).

## Additional information

### Funding

| Funder | Grant reference number | Author |
|---|---|---|
| Ministerio de Economía y Competitividad | PCIN-2013-018-C03-02 | Jesús Giraldo |
| Ministerio de Economía y Competitividad | SAF2014-58396-R | Jesús Giraldo |
| Fondation pour la Recherche Médicale | DEQ20130326522 | Jean-Philippe Pin |
| European Commission | Neuron eranet, ANR-12-NEUR-0003-05 | Cyril Goudet |
| Agence Nationale de la Recherche | ANR-12-NEUR-0003 | Cyril Goudet |
| Ministerio de Economía y Competitividad | CTQ2013-48767-C3-1-R | Santi Nonell |
| Ministerio de Economía y Competitividad | CTQ2015-71896-REDT | Santi Nonell |
| Generalitat de Catalunya | 2014SGR304 | Santi Nonell |
| Ministero della Salute | ERA-NET NEURON "LIGHTGLUPAIN" | Ferdinando Nicoletti |
| Ministerio de Economía y Competitividad | PCIN-2013-017-C03-01 | Amadeu Llebaria |
| Ministerio de Economía y Competitividad | CTQ2014-57020-R | Amadeu Llebaria |

| | | |
|---|---|---|
| Generalitat de Catalunya | 2014SGR109 | Amadeu Llebaria |
| Generalitat de Catalunya | 2014CTP0002 | Amadeu Llebaria |
| Ministerio de Economía y Competitividad | SAF2014-55700-P | Francisco Ciruela |
| Instituto de Salud Carlos III | PIE14/00034 | Francisco Ciruela |
| Fundació la Marató de TV3 | Grant 20152031 | Francisco Ciruela |
| Institució Catalana de Recerca i Estudis Avançats | ICREA Academia-2010 | Francisco Ciruela |
| Agentschap voor Innovatie door Wetenschap en Technologie | SBO-140028 | Francisco Ciruela |
| Ministerio de Economía y Competitividad | PCIN-2013-019-C03-03 | Francisco Ciruela |
| Generalitat de Catalunya | 2014 SGR 1054 | Francisco Ciruela |

The funders had no role in study design, data collection and interpretation, or the decision to submit the work for publication.

## Author contributions

JF, FM, Data curation, Investigation, Methodology; ML-C, Formal analysis, Investigation, Methodology; SNot, PS, PDP, RB-O, Data curation, Formal analysis, Investigation, Methodology; GB, Conceptualization, Data curation, Supervision, Validation, Methodology, Project administration; XR, Formal analysis, Supervision, Validation, Investigation; JC, Formal analysis, Supervision, Investigation, Visualization; JG, Software, Supervision, Validation, Investigation, Visualization; J-PP, Conceptualization, Resources, Supervision, Validation, Visualization, Methodology; VF-D, Formal analysis, Supervision, Validation, Investigation, Visualization, Writing—original draft; CG, Supervision, Validation, Investigation, Visualization, Methodology; SNon, Resources, Formal analysis, Supervision, Methodology, Writing—original draft; FN, Conceptualization, Data curation, Formal analysis, Supervision, Validation, Investigation, Visualization, Writing—original draft; AL, Conceptualization, Resources, Supervision, Validation, Visualization, Methodology, Writing—original draft; FC, Conceptualization, Resources, Formal analysis, Supervision, Funding acquisition, Validation, Investigation, Visualization, Methodology, Writing—original draft, Writing—review and editing

## Author ORCIDs

Serena Notartomaso, http://orcid.org/0000-0003-4374-9233
Pamela Scarselli, http://orcid.org/0000-0002-4245-0849
Paola Di Pietro, http://orcid.org/0000-0003-1327-1961
Roger Bresolí-Obach, http://orcid.org/0000-0002-7819-7750
Giuseppe Battaglia, http://orcid.org/0000-0001-7571-3417
Xavier Rovira, http://orcid.org/0000-0002-9764-9927
Jesús Giraldo, http://orcid.org/0000-0001-7082-4695
Jean-Philippe Pin, http://orcid.org/0000-0002-1423-345X
Víctor Fernández-Dueñas, http://orcid.org/0000-0001-7834-2965
Cyril Goudet, http://orcid.org/0000-0002-8255-3535
Ferdinando Nicoletti, http://orcid.org/0000-0003-0917-443X
Amadeu Llebaria, http://orcid.org/0000-0002-8200-4827
Francisco Ciruela, http://orcid.org/0000-0003-0832-3739

## Ethics

Animal experimentation: Animals were housed and tested in compliance with the guidelines described in the Guide for the Care and Use of Laboratory Animals and following the European Union directives (2010/63/EU). All efforts were made to minimize animal suffering and the number of animals used. For the formalin animal model of pain adult male CD-1 mice (Charles River Laboratories, L'Arbresle, France) weighing 20-25 g were used and for the neuropathic pain animal model adult male C57BL/6J mice (Charles River, Calco, Italy) weighing 20-25 g were employed. All animals

were housed in groups of five in standard cages with ad-libitum access to food and water and maintained under 12 h dark/light cycle (starting at 7:30 AM), 22°C temperature, and 66% humidity (standard conditions).

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
