## [Decision Letter]

Thank you for submitting your article "Optical control of pain in vivo with a photoactive mGlu5 receptor negative allosteric modulator" for consideration by *eLife*. Your article has been reviewed by two peer reviewers, and the evaluation has been overseen by Gary Westbrook as the Senior Editor and Reviewing Editor. The following individual involved in review of your submission has agreed to reveal his identity: Robert Gereau (Reviewer #1). The reviewers have discussed the reviews with one another and the Reviewing Editor has drafted this decision to help you prepare a revised submission.

Summary:

The demonstration of effective modulation of behavior with this photocaged compound in mice is exciting pending the inclusion of appropriate controls. However, both reviewers were surprised by the lack of mention of the prior literature relevant to this topic, including the authors' own work in Nature Chemical Biology and work on optical control of nociception. This is critical to put the new results in context. As a result, the authors have in essence oversold the novelty of the study, and must discuss this compound and approach in relation to previously published work. For example, the reasons to use this photocaged compound over a photoswitchable compound must be discussed. Both reviewers also raised a number of technical concerns as indicated in their comments below. It is essential that these issues are addressed in the revised version.

*Reviewer #1:*

This is a very interesting manuscript describing the development and characterization of a photo-activated mGlu5 NAM, JN-NP-26, a derivitized version of raseglurant, a previously characterized NAM of mGlu5. The authors demonstrate photoconversion of JN to raseglurant, and the analgesic efficacy of the compound, conditionally on photoconversion to raseglurant, in both central (thalamic) and peripheral (hindpaw) modes. The compound presents a potentially very useful tool for the field. Overall – this is a very interesting and potentially important report. The manuscript would be greatly strengthened by the inclusion of some key additional experiments and controls, as detailed below:

1) Raseglurant is a systemically active mGlu5 NAM. The author demonstrates clearly photoconversion of JN-NP-26 to raseglurant, and that photoconversion of systemically applied JN-NP-26 with light applied to the thalamus or hindpaw leads to analgesia. What is not clear, is whether the effects of the converted JN-NP-26 are due to actions locally. That is – it is possible that the local photoconversion in the thalamus or hindpaw leads to systemically effective doses of raseglurant, and thus, does not indicate that they have achieved the "precise targeting in time and space" that they seek. Additional controls are necessary. What happens if the photostimulation is applied not to the hindpaw, but to (for example) the contralateral hindpaw, or back skin? In the brain – what happens if the light is applied not to the thalamus, but for example to a region of the brain not thought to be involved in pain – (who knows… cerebellum? Olfactory bulb?). This is critical for the interpretation of the data.

2) Related – the authors have the stated goal of providing mGlu5 blockade-mediated analgesia without the undesired side effects seen with systemic administration of some mGlu5 NAMs. However – they never actually show that these side effects are absent with the JN-NP-26 photo conversion.

3) Further related – the stated reason for discontinuation of raseglurant is liver tox – however, it is unlikely that liver toxicity is mGlu5 mechanism-based. Thus, it is entirely possible that JN-NP-26 may have the same issues with liver toxicity. This has not been tested here, and should be. A comparison of raseglurant to JN-NP-26 on liver tox screens.

4) There is no mention in the manuscript (unless I missed it) as to whether the experiments were performed blind to treatment. This is a necessity.

*Reviewer #2:*

In this work, Font et al. characterize a photocaged derivative of raseglurant, a negative allosteric modulator the metabotropic glutamate receptor 5 (mGluR5). The compound, JF-NP-26, can be uncaged by illuminating with 405 nm light, both in vitro and in vivo, to reveal raseglurant with spatial and temporal precision, allowing local modulation of mGluR5 receptors. The authors show that JF-NP-26 can be intraperitoneally injected and readily diffuses throughout the mouse before local irradiation at either the paw or in the brain uncages the raseglurant. In two separate in vivo mouse models for pain, hind paw formalin injection and chronic constriction injury of the sciatic nerve, significant differences in antinociceptive effect were observed between irradiated and unirradiated JF-NP-26. This is a solid paper that moves the field forward, however, I don't think it passes the significance test as currently written.

1) The Abstract claims this report to be the first example of optical control of analgesia in vivo, but that is not the case, see Mourot, A..… Kramer, R.H. "Rapid optical control of nociception with an ion channel photoswitch" Nat. Methods 2012.

2) I am also baffled by the omission of discussion and citation of some of the authors own work in the field. Two of their recent papers (Pittolo, S..… Gorostiza, P. "An allosteric modulator to control endogenous G protein-coupled receptors with light " Nat. Chembio. 2014. And Rovira, X.… LLebaria, A. " OptoGluNAM4.1, a Photoswitchable Allosteric Antagonist for Real-Time Control of mGlu4 Receptor Activity" Cell Chem. Bio. 2016) report photoswitchable mGluR4 and mGluR5 negative allosteric modulators. The authors need to make a convincing case for why they are using the single use photocage compound JF-NP-26 instead one of their reversibly activatable compounds.

3) My last major concern is with the penultimate paragraph in the Discussion, for example ("Our data indicate that thalamic mGlu5 receptors are critical for the expression of nociceptive sensitization and can be specifically targeted by analgesic drugs in the treatment of neuropathic pain"). Is it not possible/likely, that on the timescale of the formalin experiments (15 min diffusion followed by 5 min irradiation and diffusion) that uncaged raseglurant, whether originating in the foot or brain, will diffuse through the mouse at pharmacologically relevant concentrations, making it difficult to assess the impact of local uncaging?

---

## [Author Response]

*Summary:*

*The demonstration of effective modulation of behavior with this photocaged compound in mice is exciting pending the inclusion of appropriate controls. However, both reviewers were surprised by the lack of mention of the prior literature relevant to this topic, including the authors' own work in Nature Chemical Biology and work on optical control of nociception. This is critical to put the new results in context. As a result, the authors have in essence oversold the novelty of the study, and must discuss this compound and approach in relation to previously published work. For example, the reasons to use this photocaged compound over a photoswitchable compound must be discussed. Both reviewers also raised a number of technical concerns as indicated in their comments below. It is essential that these issues are addressed in the revised version.*

We are pleased to hear that our manuscript might be suitable for publication in *eLife* provided we address all reviewer’s concerns. Indeed, we found referee’s comments very appropriate and useful. Thus, we followed their indications in order to improve the quality of our work and consequently, a large amount of new experiments (mostly controls) have been done to this purpose. As stated, we performed some important controls which validate our results and help us to properly discuss within a general context. Also, we discussed our (and others) previous literature important to this topic. Overall, we thank the referees for their constructive comments which indeed forced us to improve the manuscript’s quality.

*Reviewer #1: […] 1) Raseglurant is a systemically active mGlu5 NAM. The author demonstrates clearly photoconversion of JN-NP-26 to raseglurant, and that photoconversion of systemically applied JN-NP-26 with light applied to the thalamus or hindpaw leads to analgesia. What is not clear, is whether the effects of the converted JN-NP-26 are due to actions locally. That is – it is possible that the local photoconversion in the thalamus or hindpaw leads to systemically effective doses of raseglurant, and thus, does not indicate that they have achieved the "precise targeting in time and space" that they seek. Additional controls are necessary. What happens if the photostimulation is applied not to the hindpaw, but to (for example) the contralateral hindpaw, or back skin? In the brain – what happens if the light is applied not to the thalamus, but for example to a region of the brain not thought to be involved in pain – (who knows… cerebellum? Olfactory bulb?). This is critical for the interpretation of the data.*

The referee raised an important question here which indeed is critical for the interpretation of the results. Accordingly, we performed the additional controls suggested by the referee. First, in the formalin animal model of pain we photostimulate the contralateral hind paw to determine whether the local photoconversion of raseglurant in the periphery leads to a systemic analgesic effect. Importantly, we did not observe any systemic antinociceptive effect upon contralateral hind paw irradiation These important controls are now included in the new version of the manuscript (i.e. “Interestingly, under the same experimental conditions when the contralateral hind paw was irradiated no antinociceptive effect was observed (Figure 6, middle panel, column C), thus demonstrating that the local and peripheral photoconversion of JF-NP-26 do not yield systemic raseglurant-mediated analgesia.”).

Second, we also performed another additional control consisting on the irradiation of the striatum. To this end, we implanted optic fibers bilaterally in the striatum and performed similar formalin experiments. Interestingly, the photorelease of raseglurant in the striatum do not promote any antinociceptive effect, thus validating both our light-mediated spatiotemporal manipulation of central mGlu5 receptors and the role of these receptors at the pain neuraxis. Indeed, these results are included and discussed in the new version of the manuscript (i.e. “Importantly, under the same experimental conditions when the striatum was irradiated no antinociceptive effect was observed (Figure 6, lower panel, column S), thus demonstrating that the striatal photoconversion of JF-NP-26 do not preclude pain transmission.”).

*2) Related – the authors have the stated goal of providing mGlu5 blockade-mediated analgesia without the undesired side effects seen with systemic administration of some mGlu5 NAMs. However – they never actually show that these side effects are absent with the JN-NP-26 photo conversion.*

We agree with the referee that in our original manuscript we didn’t provide any evidence supporting the absence of raseglurant-related adverse effect in the JN-NP-26 treatment. In an attempt to address this important issue, we performed a new set experiments evaluating the impact of the systemic administration and in vivo photoactivation of JF-NP-26 in memory. The results are incorporated and discussed as follows: “On the other hand, it has been reported that mGlu5 receptor NAMs may produce behavioural undesirable side effects. […] Overall, these results demonstrated that neither the systemic administration nor the thalamic photoactivation of JF-NP-26 cause memory impairment in mice.”

*3) Further related – the stated reason for discontinuation of raseglurant is liver tox – however, it is unlikely that liver toxicity is mGlu5 mechanism-based. Thus, it is entirely possible that JN-NP-26 may have the same issues with liver toxicity. This has not been tested here, and should be. A comparison of raseglurant to JN-NP-26 on liver tox screens.*

As the referee highlighted, the main reason for raseglurant discontinuation was hepatotoxicity. Thus, despite good clinical efficacy, tolerability and safety, the long-term administration of raseglurant in migraine patients resulted in a disappointing high incidence of hepatic transaminase abnormalities, thus halting any further development of this compound. Apparently, this hepatic effect of raseglurant has been shown to be related to its metabolism and not to the mGlu_5_ receptor-mediated function (Addex data on file). Indeed, the hepatotoxicity was reported in humans and not in preclinical animal models of pain. Thus, to properly discuss this contention in our manuscript we assessed the raseglurant and JF-NP-26 liver toxicity in mice. The results are incorporated and discussed as follows: “Raseglurant has been shown to possess effectiveness on migraine treatment [Mogil, 2009]. […] Thus, these results demonstrated that raseglurant and JF-NP-26 treatments used in our pain animal models were safe as no liver toxicity was observed.”

*4) There is no mention in the manuscript (unless I missed it) as to whether the experiments were performed blind to treatment. This is a necessity.*

We apologize for forgetting this important issue. Indeed, it is a routine in our laboratory to perform animal experimentation blind to treatment. Thus, in the new version of the manuscript we clearly stated this matter (i.e. “All animal experimentation was carried out by a researcher blind to drug treatments.”)

*Reviewer #2:*

*In this work, Font et al. characterize a photocaged derivative of raseglurant, a negative allosteric modulator the metabotropic glutamate receptor 5 (mGluR5). The compound, JF-NP-26, can be uncaged by illuminating with 405 nm light, both in vitro and in vivo, to reveal raseglurant with spatial and temporal precision, allowing local modulation of mGluR5 receptors. The authors show that JF-NP-26 can be intraperitoneally injected and readily diffuses throughout the mouse before local irradiation at either the paw or in the brain uncages the raseglurant. In two separate in vivo mouse models for pain, hind paw formalin injection and chronic constriction injury of the sciatic nerve, significant differences in antinociceptive effect were observed between irradiated and unirradiated JF-NP-26. This is a solid paper that moves the field forward, however, I don't think it passes the significance test as currently written.*

*1) The Abstract claims this report to be the first example of optical control of analgesia in vivo, but that is not the case, see Mourot, A..… Kramer, R.H. "Rapid optical control of nociception with an ion channel photoswitch" Nat. Methods 2012.*

We apologize for the inconvenience. The referee is right, this is not the first example of optical control of analgesia in vivo with a drug. Thus, we amended the text accordingly in order to be more precise (i.e. “Hence, we revealed the first example of optical control of analgesia in vivo using a photocaged mGlu5 receptor negative allosteric modulator”).

*2) I am also baffled by the omission of discussion and citation of some of the authors own work in the field. Two of their recent papers (Pittolo, S..… Gorostiza, P. "An allosteric modulator to control endogenous G protein-coupled receptors with light " Nat. Chembio. 2014. And Rovira, X.… LLebaria, A. " OptoGluNAM4.1, a Photoswitchable Allosteric Antagonist for Real-Time Control of mGlu4 Receptor Activity" Cell Chem. Bio. 2016) report photoswitchable mGluR4 and mGluR5 negative allosteric modulators. The authors need to make a convincing case for why they are using the single use photocage compound JF-NP-26 instead one of their reversibly activatable compounds.*

We agree with the referee that the rationale of using a photocaged mGlu_5_ receptor NAM instead of the existing photoswitchable compound should be clearly stated early in the manuscript. Thus, in the Introduction we cite and discuss our own work regarding mGlu_5_ receptor NAM photoswitchable compounds as well as we provide a proper reasoning for using photocaged compounds (i.e. “The optical control of analgesia has been previously reported using an ion-channel photoswitch [Mourot et al., 2012]. […] Therefore, we aimed to synthesize an inactive mGlu5 receptor NAM which yields an effective analgesic drug upon illumination. An effective strategy….”).

*3) My last major concern is with the penultimate paragraph in the Discussion, for example ("Our data indicate that thalamic mGlu5 receptors are critical for the expression of nociceptive sensitization and can be specifically targeted by analgesic drugs in the treatment of neuropathic pain"). Is it not possible/likely, that on the timescale of the formalin experiments (15 min diffusion followed by 5 min irradiation and diffusion) that uncaged raseglurant, whether originating in the foot or brain, will diffuse through the mouse at pharmacologically relevant concentrations, making it difficult to assess the impact of local uncaging?*

The referee raised an important question here, namely the possibility that the photorelease of raseglurant either in the hind paw or in the thalamus could act systemically. We carefully addressed this issue by performing some important controls:

i) In the formalin animal model of pain we photostimulate the contralateral hind paw to determine whether the local photoconversion of raseglurant in the periphery leads to a systemic analgesic effect. Importantly, we did not observe any systemic antinociceptive effect upon contralateral hind paw irradiation.

ii) We also performed another additional control consisting of the irradiation of the striatum. To this end, we implanted optic fibers bilaterally in the striatum and performed similar formalin experiments. Interestingly, the photorelease of raseglurant in the striatum do not promote any antinociceptive effect, thus validating both our light-mediated spatiotemporal manipulation of central mGlu_5_ receptors and the role of these receptors at the pain neuraxis.

Indeed, these important controls are now included in the new version of the manuscript.